# Multi-Degree-of-Freedom for Underwater Optical Wireless Communication with Improved Transmission Performance

Anliang Liu [1] , Ruolin Zhang [1], Bin Lin [1,2,*] and Hongxi Yin [3]

1   School of Information Science and Technology, Dalian Maritime University, Dalian 116026, China
2   Peng Cheng Laboratory, Shenzhen 518052, China
3   School of Information and Communication Engineering, Dalian University of Technology,
    Dalian 116024, China
*   Correspondence: binlin@dlmu.edu.cn

**Abstract:** Underwater optical wireless communication (UOWC) has great potential to provide high-speed and intensive communications over short ranges underwater. However, the mobility of the UOWC system is limited by the strict alignment requirements between the transceivers. In this paper, a multi-degree-of-freedom (MDOF) UOWC system with high flexibility and improved transmission performance is proposed and experimentally demonstrated based on the off-the-shelf light-emitting diode (LED) source. A hardware pre-equalization circuit is employed at the transmitter to extend the modulation bandwidth from 5.03 MHz to 50 MHz. At the receiving end, a Fresnel lens array is constructed to achieve efficient convergence of multiple incident optical signals from different directions. To improve the underwater signal transmission quality, we designed an additional digital signal recovery module after the trans-impedance amplifier. Finally, an experimental system is established with a 460 nm blue LED. The communication reliability of the system is verified by the measurement of the eye diagram and the bit error rate of the recovered signal at the receiving end. The experimental results show that optical signals from three different incident directions with a maximum data rate of 100 Mbps are reliably transmitted over a 1.2-m-long water tank using the non-return-to-zero on-off-keying modulation format.

**Keywords:** underwater optical wireless communication; multi-degree-of-freedom; light-emitting diode; pre-equalization; lens array

## 1. Introduction

The oceans cover three-quarters of the earth's surface and possess abundant resources. Especially with the scarcity of terrestrial resources, marine development has been attracting emerging attention in recent years [1,2]. Underwater wireless communication (UWC) technology plays a significant role in marine resource exploration, aquaculture development, military defense tasks, ecosystem monitoring, and maritime search and rescue [3,4]. UWC refers to the method of data transmission in the unguided water medium. The most commonly used methods are acoustic communication, underwater electromagnetic wave communication, and underwater optical wireless communication (UOWC) [5]. Compared with the other two technologies, the UOWC technology has apparent advantages in data rate, transmission delay, device size, and power consumption, making it an attractive and practical solution for short- to medium-distance and high-speed communication in underwater environments [6,7].

The blue-green wavelength (450–550 nm) light has a window of low absorption in seawater, which is the premise of the UOWC system application [8]. Laser diodes (LDs) and light-emitting diodes (LEDs) are commonly utilized as light sources in typical UOWC systems. In general, the LD light source is more suitable for high-speed and long-distance transmission due to its excellent collimation and high modulation bandwidth. However,

the acquisition and tracking issues of the laser beams will limit the mobility of the UOWC system in some scenarios, such as regional communication applications for autonomous underwater vehicles (AUVs) [9,10]. For example, when AUVs collect data from underwater fixed sensor nodes, the precise alignment increases the burden on the AUV's navigation and attitude control systems. Especially for AUV swarm application scenarios, as shown in Figure 1, when they accomplish collective tasks, such as data collection from sensor networks or seabed mapping, a high-mobility and wide-coverage UOWC system is urgently needed [11,12]. Moreover, for the real-time ocean observing system, the UOWC also plays an important role in information relay in natural hazards monitoring and the collection of biogeochemical parameters for seawater [13].

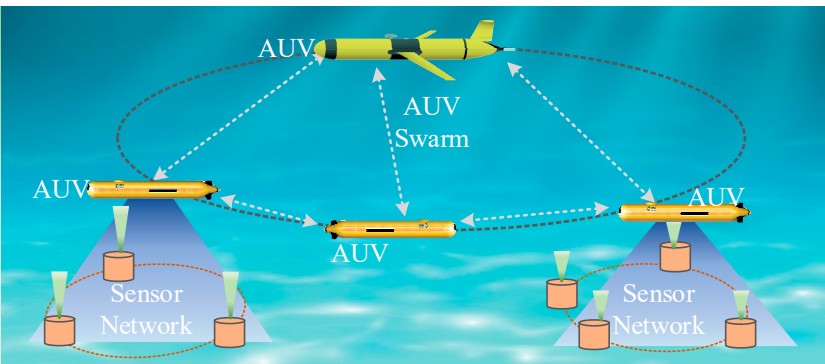

**Figure 1.** Conception of AUV swarm collaborative operation.

In comparison, the LED light source has a larger divergence angle with reduced alignment requirements, which is more suitable for underwater mobile networks and has been widely researched in recent years. In addition, the LED provides significant advantages in terms of power consumption, temperature sensitivity, system costs, and easy compatibility with underwater lighting equipment [14]. The Woods Hole Oceanographic Institution of the United States of America developed a low-power UOWC system with LEDs and the IrDA protocol to achieve a 5 m transmission in the seawater, proving the feasibility of LEDs in underwater communication systems [15]. Doniec et al. designed two prototypes of AquaOptical I and AquaOptical II. A bidirectional transmission with a data rate of several Mbps based on LEDs was achieved over a transmission distance of 50 m [16,17]. LEDs have also been used in commercial underwater modems, such as the Sonardyne BlueComm, which can operate over 150 m at a maximum rate of 10 Mbps [11,18]. However, the modulation bandwidth of the traditional commercial LED source is only approximately 3 MHz due to its large RC time constant, which limits the data rates of the system [19]. Several approaches have been explored to improve the −3 dB modulation bandwidth and realize faster data transmission [20]. The equalization technique is one of the bandwidth enhancement approaches. In [21], the −3 dB modulation bandwidth of the proposed UOWC system was extended over 700 MHz by utilizing a hardware single-stage equalizer. With a cascaded pre-equalization circuit, a bandwidth of 366 MHz and data rates of 1.3 Gbps were achieved [22]. By employing high -efficiency modulation technologies, such as orthogonal frequency division multiplexing (OFDM) or discrete multi-tone (DMT), data rates can also be effectively increased [23]. Xu et al. adopted the OFDM technology with 64-quadrature amplitude modulation (QAM) to realize 127.07 Mbps of offline communication over a 2-m underwater channel [24]. Another approach is to decrease the area size of the LEDs by using micro-LEDs as the light source. In [25], Tian et al. utilized a GaN-based micro-LED to establish a UOWC link with a data rate of 800 Mbps over a 0.6-m length, and the used micro-LED had a maximum modulation bandwidth of 160 MHz. However, the optical output power of the micro-LED is insufficient for long-distance transmission. Tsai et al. proposed a 4 × 4 LED array to increase the output power and realize underwater data rates up to 200 Mbps [26]. Although the transmission data rates

of the LEDs have been significantly improved, considering the underwater attenuation effect, most UOWC systems need to use a Fresnel lens at the receiving end to improve the receiver's sensitivity for a longer transmission distance. However, the convergence ability of the lens is sensitive to the incident angle of the received optical signal, which means that the alignment is still a restriction [27]. Especially for the optical communication links between the AUVs, a transceiver alignment system with high dynamics is essential to alleviate the influence of underwater waves on the attitude of the AUVs. Hence, there are still challenges in making full use of the large divergence angle of the LED to enhance the mobility of the UOWC systems.

In this paper, we propose and experimentally demonstrate a multi-degree-of-freedom (MDOF) UOWC system to achieve higher flexibility based on the off-the-shelf LED source. Firstly, a Fresnel lens array is constructed at the receiver terminal to enhance the convergence and sensitivity of the received optical signals with different incident angles. Then, we design an additional digital signal recovery module (DSR) after the traditional transimpedance amplifier (TIA) to improve the underwater link impairments introduced during signal transmission, including attenuation and timing jitter. Furthermore, we employ a pre-equalization circuit at the transmitter terminal to enhance the −3 dB bandwidth of commercial off-the-shelf LEDs, which enables the proposed system to achieve high-speed data transmission. Moreover, the cost, power consumption, and complexity of the system can be reduced by its practical compatibility with underwater lighting equipment. Finally, we establish an experimental verification system with a 100-Mbps data signal using the non-return-to-zero on-off-keying (NRZ-OOK) modulation format over a 1.2-m water tank. The transmission reliability for different incident signals is demonstrated by the experimental results of the eye diagrams and BER curves.

## 2. System Design

### 2.1. Fresnel Lens Array

At the receiver end of the traditional UOWC system, the Fresnel lens is often utilized in front of the photodetector to converge the received optical signal, as shown in Figure 2a. However, the focal point location of the Fresnel lens will change with the receiver's movement, which results in the degradation of the communication performance, and even the interruption of the data link. The alignment requirements are essential for the UOWC system, which results in restricted system mobility. After the transceiver is aligned, the receiver can only move in the horizontal direction to ensure the stability of the communication link, which means the system has one DOF in the horizontal direction. When the optical signal is incident horizontally, the received power of the signal after being transmitted through the water tank can be expressed as [28]:

$$P_r(d) = P_t \times \eta_t \times \eta_r \times \frac{A_r}{\pi \left( d \tan \frac{\theta}{2} \right)^2} \times \exp[-c(\lambda)d] \qquad (1)$$

where $P_t$ is the transmitting power, $\eta_t$ is the optical efficiency of the transmitter, $\eta_r$ is the optical efficiency of the receiver, $A_r$ is the convergence area of the lens, $d$ is the transmission distance, $\theta$ is the divergence angle of the LED source, and $c(\lambda)$ is the total attenuation coefficient of water, which can be expressed as:

$$c(\lambda) = a(\lambda) + b(\lambda) \qquad (2)$$

where $a(\lambda)$ is the absorption coefficient, $b(\lambda)$ is the scattering coefficient, and the units of all the coefficients are m 1. The typical values of $a(\lambda)$, $b(\lambda)$, and $c(\lambda)$ are associated with four major water types [29].

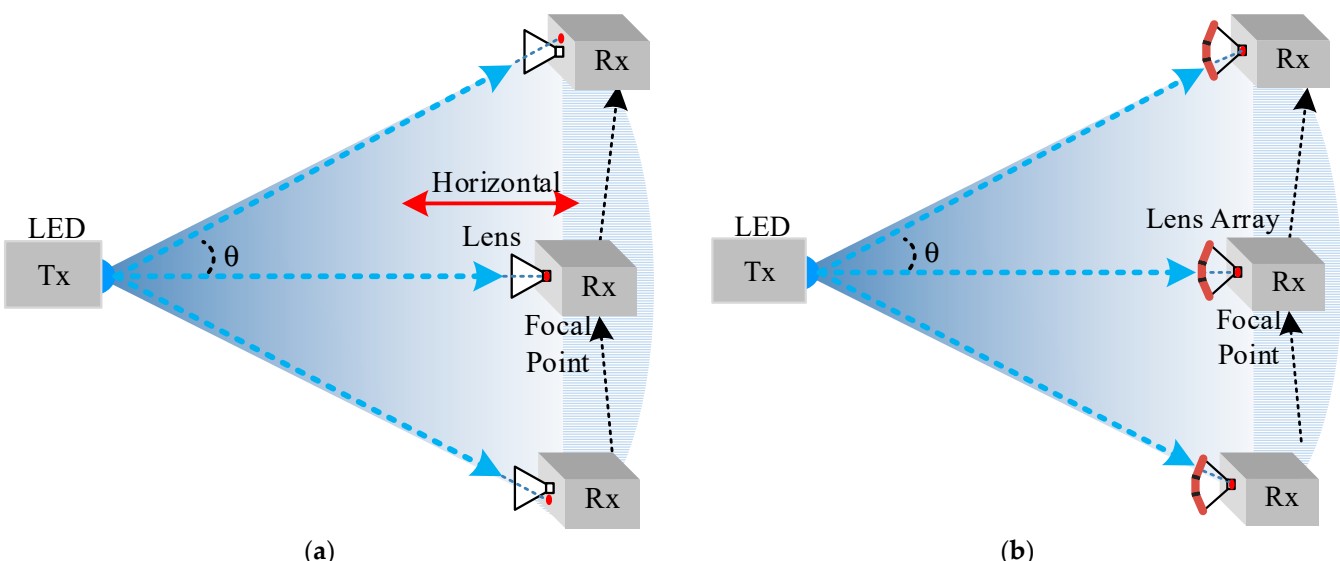

**Figure 2.** Schematic diagram of the dynamic UOWC system. (**a**) With one lens; (**b**) With lens array.

To enhance the mobility of the UOWC system, we propose a Fresnel lens array structure to realize an effective convergence of multiple-incident optical signals from different directions, as shown in Figure 2b. The DOF available for the UOWC system will increase with the number of lenses used in the array. Considering the limited space inside the receiver, the area of each Fresnel lens decreases as the number of lenses added to the array increases, which leads to a degradation of the received power. Hence, we take a 6-lens array structure as an example to verify the flexibility and communication reliability. All the lenses are combined into a hemispherical shape according to the schematic diagram shown in Figure 3a. We designed a core frame of the lens array using a 3D printer (Z600plus, HORI). A regular pentagonal Fresnel lens with a side length of 16 mm is placed in the center, and five regular hexagonal Fresnel lenses with the same side length are fixed on different sides by rotatable axes. All the lenses are produced from poly (methyl methacrylate), which has a refractive index of 1.49. The assembled 6-lens array is shown in Figure 3b. The focal length of all the lenses is set to 20 mm. The focal point of each Fresnel lens is assigned to 20 mm. By adjusting the angles of the lenses, the focal points of the incident optical signals from six different directions can be converged on the same photodetector. A positive-intrinsic-negative (PIN) photodiode (Hamamatsu S6967) with a large active photosensitive area of 26.4 mm$^2$ is employed to realize an optical-to-electro (O/E) conversion of the received signal. The −3 dB bandwidth of the PIN is 50 MHz, and the photosensitivity at 450 nm can be 0.3 A/W.

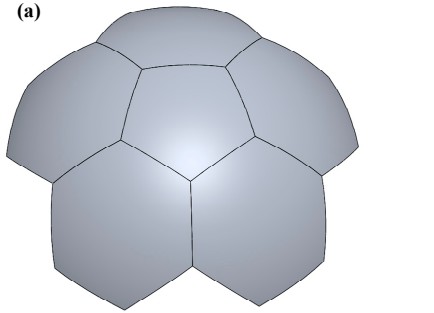
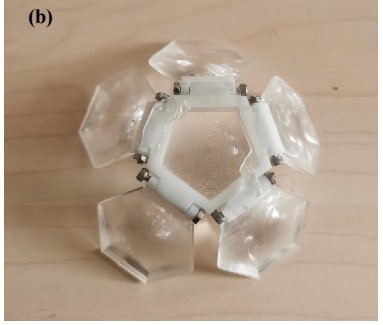

**Figure 3.** (**a**) Schematic diagram of a 6-lens array; (**b**) Assembled 6-lens array.

### 2.2. Pre-Equalization

The LED source is an essential device in underwater vehicles for video observation. To realize high-speed data transmission and effective compatibility with underwater lighting equipment, we employed a frequency domain pre-equalization circuit at the transmitter to improve the frequency response characteristics of the commercial off-the-shelf LEDs. Figure 4 shows the bridged-T pre-equalization circuit used in the proposed UOWC system. It comprises two RLC networks with the equivalent impedances of $Z_{11}$ and $Z_{22}$, respectively.

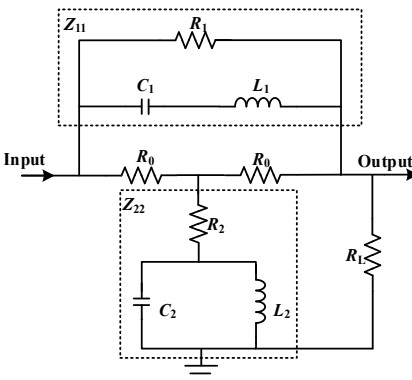

**Figure 4.** Schematic of the bridged-T pre-equalization circuit.

The impedance $Z_{11}$ consists of a resistor $R_1$, a capacitor $C_1$, and an inductor $L_1$. The impedance $Z_{22}$ consists of a resistor $R_2$ and a two-port network in series. The two-port network consists of a capacitor $C_2$ and an inductor $L_2$ in parallel. The two-series resistors $R_0$ are set to equal the load impedance $R_L$. The bridged-T pre-equalization circuit should follow the following relationship provided by [22].

$$Z_{11}Z_{22} = R_0^2 \tag{3}$$

where $Z_{11}$ and $Z_{22}$ are given by

$$Z_{11} = \frac{1}{\frac{1}{R_1} + \frac{1}{j\omega L_1 + \frac{1}{j\omega C_1}}} \tag{4}$$

$$Z_{22} = R_2 + \frac{1}{\frac{1}{j\omega L_2} + j\omega C_2} \tag{5}$$

where $\omega$ is the angular frequency of the transmitted signal. For easy realization and analysis, we set $L_1 = L_2$ and $C_1 = C_2$, and thus the forward transmission gain $S_{21}$ of the pre-equalization circuit can be expressed as:

$$S_{21} = \frac{1}{1 + \frac{R_L}{R_2 + \frac{j\omega L_1}{1 - \omega^2 C_1 L_1}}} \tag{6}$$

From (6), when $(1 - \omega^2 C_1 L_1)$ tends to zero, $S_{21}$ is equal to 1. Therefore, the bandwidth $f_c$ of the pre-equalization circuit can be calculated by:

$$f_c = \frac{1}{2\pi\sqrt{L_1 C_1}} \tag{7}$$

Based on the bandwidth of the existing components in the proposed UOWC system, the parameters used for the bridged-T pre-equalization circuit are $R_1$ = 249 $\Omega$, $R_0$ = 49.9 $\Omega$, $R_2$ = 10 $\Omega$, $C_1 = C_2$ = 47 pF, and $L_1 = L_2$ = 120 nH. The S-parameters are measured by a

vector network analyzer (VNA, Keysight E5063A), operating at a frequency from 100 kHz to 18 GHz.

Figure 5 shows the forward transmission gain $S_{21}$ of the designed pre-equalizer, which has a frequency response range of 28.47 MHz to 67.08 MHz.

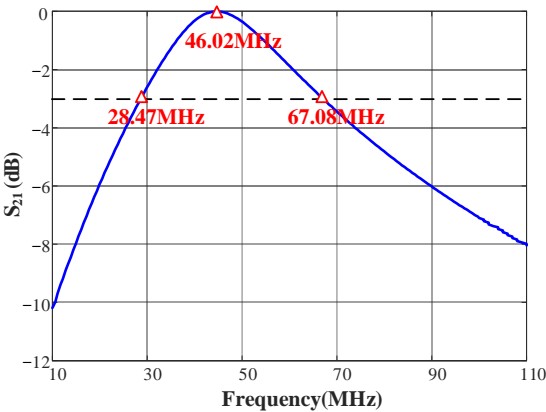

**Figure 5.** $S_{21}$ parameter of the pre-equalization circuit.

### 2.3. Digital Signal Recovery

After the TIA in the traditional receiver, we designed an additional DSR module to improve the underwater link impairments introduced during signal transmission, including attenuation and timing jitter. The DSR module consists of a limiting amplifier (LA), a phased-locked loop (PLL) circuit, and a delay-locked loop (DLL) circuit, whose schematic is shown in Figure 6. It can provide the receiver with functions of limiting amplification, clocking, and data recovery for wideband continuous data rates without the need for an external reference clock.

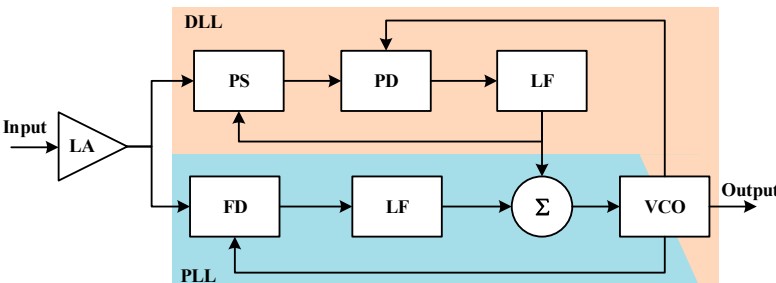

**Figure 6.** Schematic of the DSR module.

The input port is terminated with 50 Ω to the supply voltage, and the sensitivity of the LA is as low as 3.3 mV of the peak-to-peak voltage. Then, the phase of the amplified input data signal is tracked by two separate feedback loops, which share a common voltage-controlled oscillator (VCO). The PLL tracks the low-frequency components of the input jitter, and the DLL tracks the high-frequency components of the input jitter. At low jitter frequencies, a frequency detector (FD) compares the incoming data frequency to this VCO frequency. A loop filter (LF) provides a high gain to track large jitter amplitudes with small phase errors. The jitter tolerance of this PLL is roughly half a period. At higher jitter frequencies, the gain and tuning range of the VCO are not large enough to track the input jitter. In this case, the VCO control voltage becomes large and saturated. A phase shifter (PS) in the DLL circuit begins to track the input jitter. Then, a phase detector (PD) drives the VCO to a higher frequency. The gain of the loop integrator is small for high jitter frequencies. In this case, jitter accommodation is determined by the eye-opening of the input data. The PLL and DLL simultaneously provide wideband jitter tolerance and narrow-band jitter filtering. The VCO frequency is reset to 10 MHz, which is the bottom of

its range. Initially, the VCO frequency is incremented in large steps to aid fast acquisition. As the VCO frequency approaches the data frequency, the step size is reduced until the VCO frequency is within 250 ppm of the data frequency. Once the input frequency error exceeds 1000 ppm, the control returns to the frequency loop, which begins a new frequency acquisition.

## 3. Experimental Results

### 3.1. Experimental Setting

Figure 7 shows the experimental setup of the proposed MDOF UOWC system based on the Fresnel lens array, hardware pre-equalizer, and DSR module as mentioned above. At the transmitter end, the original data signal is generated by a BER tester (BERT, Anritsu MP1632C). The operating voltage for the LED is provided by a direct current (DC) power supply, and the output power of the transmitter is controlled by adjusting the DC voltage at the transmitter end. After being equalized by the bridged-T circuit, the data signal and the DC voltage signal are combined into the LED via a bias-T circuit. A 450 nm commercial blue LED with a 60° initial divergence angle is used as the light source. To realize a long-distance transmission of the transmitted optical signals, a focusing lens is placed before the water tank with a size of 1.2 m × 0.5 m × 0.5 m. The water tank is filled with fresh tap water.

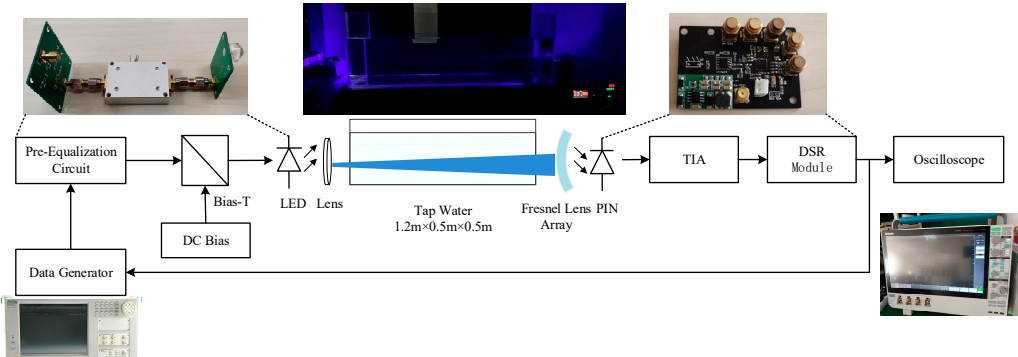

**Figure 7.** Experimental setup of the proposed MDOF UOWC system.

At the receiver side, the constructed 6-Fresnel lens array is located in front of the PIN-PD to obtain high sensitivities for incident optical signals from different angles. After the O/E conversion, the detected current signal from the PIN is amplified by a low-noise TIA (AD8001) with a flat gain bandwidth of up to 60 MHz. At the end of the receiver, the DSR module is employed to improve the quality of the received signal. Finally, the eye diagram and the BER performance of the recovered data signal can be analyzed by an oscilloscope (OSC, Tektronix MSO54) and a BERT to verify the communication reliability of the proposed UOWC system.

### 3.2. Results and Analysis

We measure the $S_{21}$ parameters of the proposed UOWC system with and without the pre-equalizer based on the commercial LED source, respectively. The measurement results are shown in Figure 8. The −3 dB bandwidth is improved from 5.03 MHz to 50 MHz after the pre-equalization. However, due to the addition of the pre-equalizer, the forward transmission gain of the system is reduced by approximately 20 dB. Different waveforms of the original signal and the equalized signal at a data rate of 100 Mbps are observed via the OSC, as shown in Figure 9. The equalized signal produces an evident overshoot on each rising and falling edge. The root-mean-square voltage is reduced from 480 mV to 280 mV, which means a higher transmitting power is required due to the additional pre-equalization circuit.

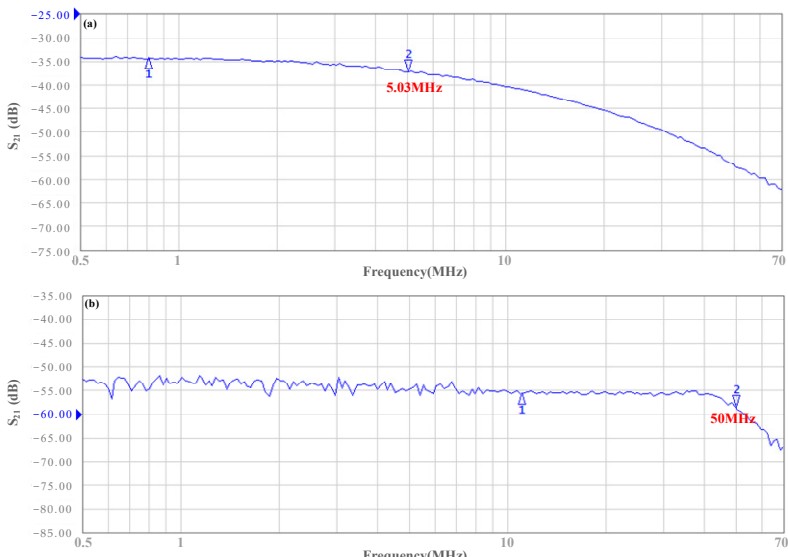

**Figure 8.** $S_{21}$ parameter of the proposed UOWC system. (**a**) Without pre-equalizer; (**b**) with pre-equalizer.

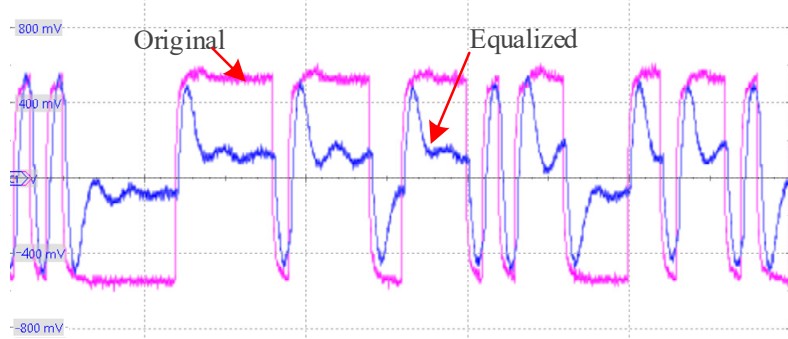

**Figure 9.** 100 Mbps signal with and without pre-equalization.

Furthermore, we test the transmission performance of the proposed MDOF UOWC system. A pseudo-random binary sequence (PRBS) with a word length of $2^9 - 1$ is generated by the BERT. The initial DC voltage is set to 3 V and the operating current of the 450 nm blue LED is 98 mA. A 30° condenser lens is fixed upon the LED to converge the transmitted signal. By adjusting the position of the light source, the received signal is incident vertically on the central plane of the Fresnel lens array. After being transmitted through the water tank, the quality of the received signal can be observed by an OSC. Figure 10 shows the eye diagrams at different data rates after being amplified by the TIA circuit. As the transmission data rate increases, the eye diagram begins to close gradually and the eye height decreases from 14.1 mV to 4.6 mV. At 100 Mbps, the peak-to-peak voltage of the received signal is only approximately 38 mV. Then, the eye diagrams processed by the DSR module are measured, as shown in Figure 11. It can be seen that the quality of the received electrical signal has been significantly optimized, and the amplitude of the output signal is amplified to 380 mV. Moreover, the total power consumption of the receiver is as low as approximately 0.55 W.

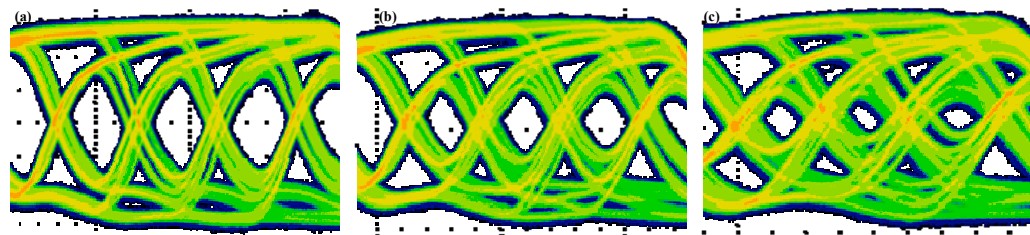

**Figure 10.** Eye diagrams of received signals at different data rates after TIA circuit. (**a**) 60 Mbps; (**b**) 80 Mbps; (**c**) 100 Mbps.

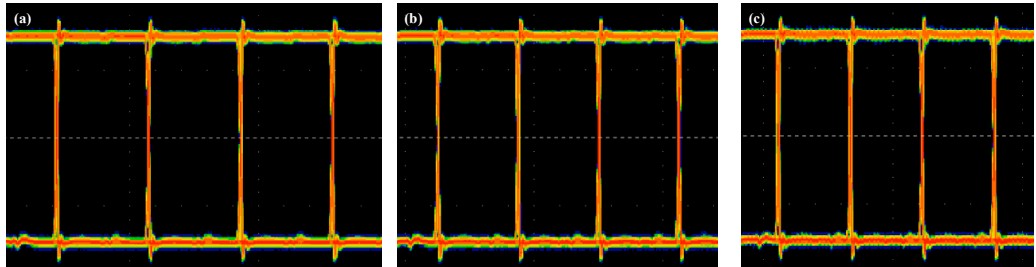

**Figure 11.** Eye diagrams after DSR. (**a**) 60 Mbps; (**b**) 800 Mbps; (**c**) 100 Mbps.

Finally, to verify the communication reliability of the proposed system under MDOF conditions, we fix the receiver position. To simulate the reception of signals with different incident angles, the light source at the transmitter is moved in parallel, on the other side of the water tank. The incident angle is defined as the angle to the normal direction of the receiver. According to the existing experimental conditions, we choose two incident angles of approximately $\pm 45°$ from the right and left sides of the receiver, as shown in Figure 12. By adjusting the receiving angle on the side of the Fresnel lens array, the received signal can be converged on the effective detection area of the same PIN detector. The BER performance of each communication link is analyzed via the BERT. Figure 12 shows the BER curves versus the different DC voltages at the transmitter. It can be seen that when the DC bias voltage is 2.74 V, the BER performance of the system is lower than $3.8 \times 10^{-3}$. As the DC voltage increases to 2.83 V, the BERs of all the data signals with different incident angles are lower than $10^{-8}$. All the received signals with different incident angles are transmitted successfully. The Fresnel lens array structure used in the experiment can effectively focus the signals emitted from different directions through the water tank channel.

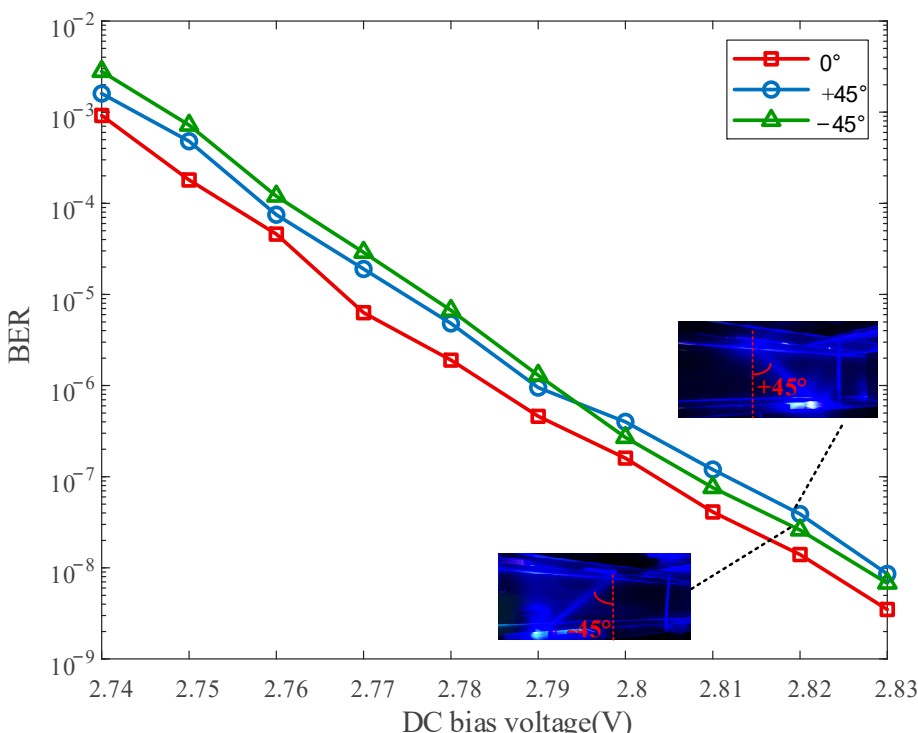

**Figure 12.** BER performance of the received signals at different DC bias voltages.

## 4. Conclusions

In this paper, a high-mobility MDOF UOWC system based on the off-the-shelf LED is proposed, and the system performance is verified by our experiments. At the transmitting end, the modulation bandwidth of the off-the-shelf LED is extended from 5.03 MHz to 50 MHz using a time-domain hardware pre-equalizer. At the receiving end, the optical signals from different angles are effectively converged through a combined Fresnel lens array. To improve the quality of the received signal, a DSR module is designed after the traditional TIA circuit. After a 100-Mbps NRZ-OOK signal is transmitted through a 1.2-m long water channel, the verification system of 0° and ±45° incident angles is demonstrated. Using the test results of the eye diagrams and BER curves at the receiving end of the system, the reliability of the MDOF system is verified. The proposed system can be used to improve the flexibility and mobility of the communication system between underwater vehicles and fully utilize the large coverage characteristics of LED light sources, which can also be deployed as part of the real-time ocean observing system to provide an efficient transmission relay link.

**Author Contributions:** Conceptualization, A.L.; formal analysis, A.L. and R.Z.; methodology, A.L.; software, A.L. and R.Z.; validation, A.L., B.L. and H.Y.; writing—original draft, A.L.; writing—review and editing, H.Y. and B.L.; visualization, A.L. and R.Z.; supervision, A.L.; project administration, A.L. and B.L.; funding acquisition, A.L. and B.L. All authors have read and agreed to the published version of the manuscript.

**Funding:** This work is supported in part by the China Postdoctoral Science Foundation under Grant 2019M651095, the Fundamental Research Funds for the Central Universities under Grant 3132022233, the National Key Research and Development Program of China under Grant 2019YFE0111600, Liaoning Revitalization Talents Program under Grant XLYC2002078, and the Major Key Project of PCL (PCL2021A03-1).

**Institutional Review Board Statement:** Not applicable.

**Informed Consent Statement:** Not applicable.

**Data Availability Statement:** The data presented in this paper are available after contacting the corresponding author.

**Acknowledgments:** The authors would like to thank the anonymous reviewers for their careful reading and valuable comments.

**Conflicts of Interest:** The authors declare no conflict of interest.

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
