# Peer review of "Multi-Degree-of-Freedom for Underwater Optical Wireless Communication with Improved Transmission Performance"

_jmse, doi:10.3390/jmse11010048_

Round 1
Reviewer 1 Report
This paper studies a multi-degree-of-freedom based on multiple incident optical signals from different directions for underwater wireless optical communication systems. The authors designed an experimental system approach with a 460nm blue light-emitting-diode (LED) to validate the communication performance reliability. These tests were done using the eye diagram measurement and the bit-error rate performance (BER). The paper is interesting, and the claims are sound. However, many references deal with the same problem of building multiple incidents and receivers’ optical signals to enhance optical wireless communication, specifically underwater optical communications.
This is an insightful work, but the current version needs significant revision.
Comments on the paper are listed below.
1. The novelties of this paper could be clearer. It is hard to grasp something in the introduction. The contribution of this paper may be more oriented toward building a setup from a practical level.
2. The principle of building multiple incidents and receivers’ optical signals, called a multi-degree of freedom UOWC, to enhance optical wireless communication to improve optical communication, specifically underwater optical communication, is not novel.
3. Please highlight the main difference between the proposed work multi-degree-of-freedom based UOWC compared to the following references [1] and [2]
[1] J. A. Simpson, B. L. Hughes, J. F. Muth, “Underwater free-space optical communication using smart transmitters and receivers, IEEE Journal on selected areas in communications, vol. 30, 2012.
[2] C. Shen, Y. Guo, and al., “20-meter underwater wireless optical communication link with 1.5 Gbps data rate”, Optical Express, vol. 24, 2016.
[3] H. M. Oubei et al. “2.3 Gbit/s underwater wireless optical communications using directly modulated 520 nm laser diode”, Optical Express, vol. 23, pp. 20743-20748, 2015.
4. The authors should check the references ([1], [2], and references that are included in [1] and [2]) I included to support their findings better and highlight more on what enhances the transmission performance from the digital signal recovery module, which plays the role of an amplifier.
5. The experimental setup of the proposed UOWC is done from a horizontal level which does not create any challenges. Further, the distance of the transmission is very short, 1.2 meters. How can we rely on the performance within this short distance? The authors should clarify this point.
6. What kind of water is used to perform the experiments? It is crucial to mention the quality of the water, clear water, turbidity harbor, etc. Please, the authors should clarify.
7. Do the authors test the performances under water quality turbidities (clear, turbid harbor, and coastal waters)? The authors should clarify.
Author Response
Manuscript ID: jmse-2040490
Title: Multi-Degree-of-Freedom for Underwater Optical Wireless Communication with Improved Transmission Performance
Authors: Anliang Liu, Ruolin Zhang, Bin Lin *, Hongxi Yin
Dear Editor,
Thanks very much for your time in editing our manuscript. We also appreciate the anonymous reviewers for the time and efforts devoting to reviewing the paper. We have revised our manuscript according to the reviewers’ suggestions and comments. Here we list our response to the reviewers. All the corrections have been marked up by the “Track Changes” function in the revised manuscript.
Response to Reviewer 1:
Comments on the paper are listed below.
1. The novelties of this paper could be clearer. It is hard to grasp something in the introduction. The contribution of this paper may be more oriented toward building a setup from a practical level.
Reply: Thanks very much for the reviewer's suggestions. To express the novelties of the manuscript more clearly, we have modified the last paragraph of the introduction in the revised manuscript.
2. The principle of building multiple incidents and receivers’ optical signals, called a multi-degree of freedom UOWC, to enhance optical wireless communication to improve optical communication, specifically underwater optical communication, is not novel.
Reply: Thanks very much for the reviewer's comments. Actually, our main findings in this paper can be summarized as follows: (1) UOWC technology has great potential to provide high-speed and intensive communications over short-ranges underwater. However, the mobility of the UOWC system is limited by the strict alignment requirements between the transceivers. In this paper, we propose a multi-degree-of-freedom UOWC system based on a Fresnel lens array structure to improve system flexibility. The proposed system is more suitable for communication between underwater agents for dynamic operations.
(2) We chose a detector with a larger area to improve the ability to capture signals at different incident angles. However, as the receiving area of the detector increases, the underwater optical signal transmission performance is aggravated by dispersion. To improve the transmission reliability of the proposed MDOF UOWC system, we designed an additional digital signal recovery module to eliminate the underwater link impairments introduced during signal transmission, including attenuation and timing jitter.
(3) Considering the system power consumption and compatibility, we employ a pre-equalization circuit to enhance the −3dB bandwidth of commercial off-the-shelf LEDs, which enables the proposed system can achieve high-speed data transmission and compatibility with underwater lighting devices. The cost, power consumption, and complexity of the system can be reduced by practical compatibility with underwater lighting equipment. Considering that the AUV is very sensitive to parameters of power consumption and system size, the above advantages are of great significance.
3. Please highlight the main difference between the proposed work multi-degree-of-freedom based UOWC compared to the following references [1] and [2]
[1] J. A. Simpson, B. L. Hughes, J. F. Muth, “Underwater free-space optical communication using smart transmitters and receivers, IEEE Journal on selected areas in communications, vol. 30, 2012.
[2] C. Shen, Y. Guo, and al., “20-meter underwater wireless optical communication link with 1.5 Gbps data rate”, Optical Express, vol. 24, 2016.
[3] H. M. Oubei et al. “2.3 Gbit/s underwater wireless optical communications using directly modulated 520 nm laser diode”, Optical Express, vol. 23, pp. 20743-20748, 2015.
Reply: We think that there are several differences between our manuscript and the reference [1]. Firstly, we use only one detector to realize the reception of the signals with different incident angles, while seven detectors were used in the reference [1]. Secondly, to improve the transmission reliability of the proposed MDOF UOWC system, we designed an additional digital signal recovery module to eliminate the underwater link impairments introduced during signal transmission, including attenuation and timing jitter. Finally, the proposed Fresnel lens array in our paper is more flexible. However, the adjustment can only be realized manually in the current system. In the following research, we will explore an automatic control scheme for the lens array.
Signal transmission up to Gbps data rate has been successfully realized in reference [2] and [3] using an LD light source. However, the UOWC system with the LD is more restrictive with the alignment requirement because of the narrow divergence angle, which is difficult to be applied to some high dynamic underwater scenarios. Therefore, the data rate is actually sacrificed in exchange for higher system dynamics and flexibility in the proposed MDOF UOWC system. Based on this requirement, we further design a pre-equalizer and DSR modules to improve the transmission data rate and performance of the system as much as possible.
4. The authors should check the references ([1], [2], and references that are included in [1] and [2]) I included to support their findings better and highlight more on what enhances the transmission performance from the digital signal recovery module, which plays the role of an amplifier.
Reply: Thanks for the reviewer's reminder. At the receiver terminal, we chose a detector with a larger area to improve the ability to capture signals at different incident angles. However, as the receiving area of the detector increases, the underwater optical signal transmission performance is aggravated by dispersion. To improve the transmission reliability of the proposed MDOF UOWC system, we designed an additional digital signal recovery module to eliminate the underwater link impairments introduced during signal transmission, including attenuation and timing jitter.
5. The experimental setup of the proposed UOWC is done from a horizontal level which does not create any challenges. Further, the distance of the transmission is very short, 1.2 meters. How can we rely on the performance within this short distance? The authors should clarify this point.
Reply: Thanks for the reviewer's reminder. Although our lens array is constructed by 6 Fresnel lenses. We only tested the transmission performance of three lenses from a horizontal level by moving the position of the water tank. Limited by the experimental conditions, the water tank used in our experimental setup is only 1.2m×0.5m×0.5m. The measurements of transmission performance for the incident signals from a vertical level may needs a deeper water tank. Similarly, the transmission distance of the system is also limited to 1.2m. However, as we know, the transmission distance of the underwater optical signal is related to the transmission power, so we analyzed the transmission performance of the system at different DC bias voltages. For a longer transmission distance, a higher transmitter power is essential. In the next step, we plan to waterproof the system and further test system performance in a swimming pool. Then, the transmission performance for all the degree of freedom under different distances can be obtained in detail.
6. What kind of water is used to perform the experiments? It is crucial to mention the quality of the water, clear water, turbidity harbor, etc. Please, the authors should clarify.
Reply: Thanks for the reviewer’s comments. In our experiment, the water tank is filled with the fresh tap water. We have included the clarification in section 3.1 of the revised manuscript.
7. Do the authors test the performances under water quality turbidities (clear, turbid harbor, and coastal waters)? The authors should clarify.
Reply: In the current experimental system, the results were measured under the fresh tap water. Thanks for the reviewer’s reminding. We plan to simulate different water quality conditions by mixing the Mg(OH)2 and fresh tap water for further measurements.

Reviewer 2 Report
This manuscript describes the performance of UWOC under blue-LED illumination, a custom-made Fresnel lens array structure, and DSR module.
In general, this manuscript is well-written, very detail, and consists in-depth and diverse experiments. However, the manuscript is not coherent enough, in a sense that the main issue in the manuscript is the design of the Fresnel lens array, but its performances are hardly tested.
Most of the results (Figs. 8 - 11) describe the performance of the UWOC w/o TIA/pre-equalization or at different data-rates (If I am mistaken, than I suggest that the authors will stress that the performances were tested w/o the Fresnel array).
Furthermore, the authors only tested the lens-array with 3 incident angles (0, -45 and +45 degs), so it is incorrect to use terms such as "multi-degree-of-freedom" (I would expect changing the pitch, yaw, and many other incident angles) and "mobility" (since the LED and the receiver were static and their position was not changed with time).
The reviewer has the following comments/concerns:
(1) The main issue that this manuscript tries to solve is the miss-alignment in UWOC (or pointing-error). As such, I think that the authors should elaborate more in the introduction part (with proper citations) on point-error and misalignment, especially due to water-waves or other causes.
(2) Fig. 2 is not clear. I suggest that the authors will enlarge the "lens array" image vs. "lens".
(3) I do not understand why the authors say that "Figure 5 shows forward transmission gain S21 of the designed pre-equalizer, which has a flatness frequency response range from 28.47MHz to 67.08MHz". I do not see this range is flat...
(4) What is the hexagonal Fresnel lenses made of? (i.e, what is the material, its refractive index, etc.)
(5) In the manuscript, some items are not detailed. For example, who is the manufacture of the TIA? 3D printer?
(6) How the authors changed the position of the light source and how the incident angles were determined/measured? I suggest to add an image that shows the different positions of the light sources..
(7) I do not understand Fig. 12. As I understand, the authors' goal was to show that the Fresnel lens array can help focus the light that comes from different angles. What is the relation between the array and the DC bias? Also, it would be interesting to check that the Fresnel lens still focus the light at other incident angles, not only 45 deg.
Author Response
Dear Editor,
Thanks very much for your time in editing our manuscript. We also appreciate the anonymous reviewers for the time and efforts devoting to reviewing the paper. We have revised our manuscript according to the reviewers’ suggestions and comments. Here we list our response to the reviewers. All the corrections have been marked up by the “Track Changes” function in the revised manuscript.
Response to Reviewer 2:
The reviewer has the following comments/concerns:
1. The main issue that this manuscript tries to solve is the miss-alignment in UWOC (or pointing-error). As such, I think that the authors should elaborate more in the introduction part (with proper citations) on point-error and misalignment, especially due to water-waves or other causes.
Reply: Thanks for the reviewer's comments. We have emphasized the effect of water-currents on the transceiver alignment systems, especially for the AUVs, and modified the introduction of the manuscript.
2. Fig2 is not clear. I suggest that the authors will enlarge the "lens array" image vs. "lens".
Reply: Thanks for the reviewer's comments. We have enlarged Figure 2 in the revised manuscript. The lens array in Figure 2 is a schematic diagram to illustrate the principle of multi-angle reception in the proposed UOWC system.
3. I do not understand why the authors say that "Figure 5 shows forward transmission gain S21 of the designed pre-equalizer, which has a flatness frequency response range from 28.47MHz to 67.08MHz". I do not see this range is flat...
Reply: Thanks for the reviewer's reminder. I'm sorry for the inaccurate expression of "flatness". We want to express that the forward transmission gain S21 has been expanded after the processing of the pre-equalizer. We have removed the word "flatness" in the revised manuscript.
4. What is the hexagonal Fresnel lenses made of? (i.e, what is the material, its refractive index, etc.)
Reply: The hexagonal lenses are made of poly (methyl methacrylate) also known as acrylic, which has a refractive index of 1.49. We have included the description in section 2.1 of the revised manuscript.
5. In the manuscript, some items are not detailed. For example, who is the manufacture of the TIA? 3D printer?
Reply: Thanks for the reviewer's comments. The TIA we used in our experiment is AD8001 and the 3D printer is HORI Z600plus. We have added the manufacturer information in section 2.1 of the revised manuscript.
6. How the authors changed the position of the light source and how the incident angles were determined/measured? I suggest to add an image that shows the different positions of the light sources.
Reply: Thanks for the reviewer's reminder. To verify the communication reliability of the proposed system under MDOF conditions, we fix the receiver and simulate the reception of signals with different incident angles by changing the position of the light source of the transmitter. We have added the experiment figures under different incident angles in Figure 12.
7. I do not understand Fig. 12. As I understand, the authors' goal was to show that the Fresnel lens array can help focus the light that comes from different angles. What is the relation between the array and the DC bias? Also, it would be interesting to check that the Fresnel lens still focus the light at other incident angles, not only 45 deg.
Reply: We are sorry for the misunderstanding. The power of the LED transmitter is controlled by the DC bias voltage. Actually, we want to evaluate the BER performance of the proposed UOWC system under the same transmission distance by changing the DC voltage. About the incident angles, we have conducted an experiment with 3 incident angles (0, -45 and +45 degs) as an example to verify the reliability of the proposed UOWC system. The proposed system can also support received signals with other incident angles by adjusting the angle of the side lens. However, the adjustment can only be realized manually in the current system. In the following research, we will explore an automatic control scheme for the lens array.
Round 2
Reviewer 2 Report
I would suggest that the authors will add explanation on how they (manually) changed the position of the light sources, and how they measured the angles (45,0..).
Author Response
Reply: Thanks for the reviewer's comments. To verify the communication reliability of the proposed system under MDOF conditions, we fix the receiver position. To simulate the reception of signals with different incident angles, the light source at the transmitter is moved in parallel, on the other side of the water tank. The incident angle is defined as the angle to the normal direction of the receiver. According to the existing experimental conditions, we choose two incident angles of about ±45° from the right and left sides of the receiver, as shown in Figure 12. The incident angles are estimated by the angle between the incident light and the normal of the receiving surface. We have modified Figure 12 to clearly express the incident angles under the measurement and added the explanation in section 3.2 of the revised manuscript.